# Modelling Working Memory using Deep Recurrent Reinforcement Learning

**Pravish Sainath**[1,2,3]          **Pierre Bellec**[1,3]

**Guillaume Lajoie** [2,3]

[1]Centre de Recherche, Institut Universitaire de Gériatrie de Montréal
[2]Mila - Quebec AI Institute
[3]Université de Montréal

## Abstract

In cognitive systems, the role of a working memory is crucial for visual reasoning and decision making. Tremendous progress has been made in understanding the mechanisms of the human/animal working memory, as well as in formulating different frameworks of artificial neural networks. In the case of humans, the visual working memory (VWM) task [1] is a standard one in which the subjects are presented with a sequence of images, each of which needs to be identified as to whether it was already seen or not. Our work is a study of multiple ways to learn a working memory model using recurrent neural networks that learn to remember input images across timesteps in supervised and reinforcement learning settings. The reinforcement learning setting is inspired by the popular view in Neuroscience that the working memory in the prefrontal cortex is modulated by a dopaminergic mechanism. We consider the VWM task as an environment that rewards the agent when it remembers past information and penalizes it for forgetting. We quantitatively estimate the performance of these models on sequences of images from a standard image dataset (CIFAR-100 [2]) and their ability to remember and recall. Based on our analysis, we establish that a gated recurrent neural network model with long short-term memory units trained using reinforcement learning is powerful and more efficient in temporally consolidating the input spatial information. This work is an initial analysis as a part of our ultimate goal to model the behavior and information processing of the working memory of the brain and to use brain imaging data captured from human subjects during the VWM cognitive task to understand various memory mechanisms of the brain.

## 1 Introduction

Memory is an essential component for solving many tasks intelligently. Most sequential tasks involve the need for a mechanism to maintain a *context*. In the brain, working memory serves as a work space to encode and maintain the most relevant information over a short period of time, in order to use it to guide behavior for cognitive tasks. Several cognitive tasks have been proposed in the Neuropsychology literature to study and understand the working memory in animals. The Visual Working Memory Task (VWM task) [1] or the classic N-back task is one of the most simple and popular ones. It involves showing sequences of images to subjects and record their responses indicating whether they have seen the image already.

On the other hand, with artificial intelligence systems, there has been very good progress in models that learn from sequences of inputs using artificial neural networks as memory for all types of learning

34 (supervised, unsupervised and reinforcement). We intend to use these developments as an ideal
35 opportunity for synergy to computationally model the working memory system of the brain.

36 As memory is an important aspect of both artificial intelligence and neuroscience, there are some
37 good studies that helped choose our models as discussed in Section 2.

## 2 Models

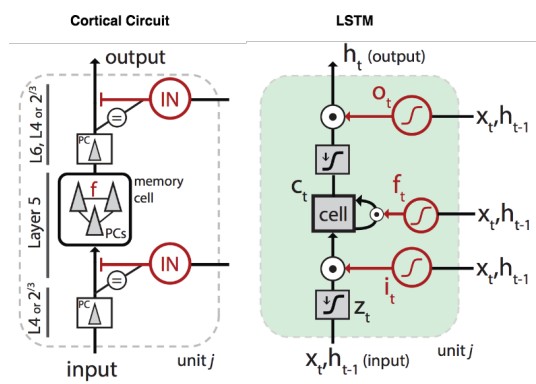
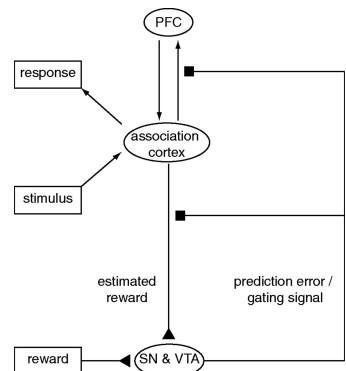

(a) Cognitive computational models of a cortical circuit (left) and working memory (right) (from [3])

(b) The brainstem dopaminergic nuclei [substantia nigra (SN) and ventral tegmental area (VTA)] interacting with memory (from [6])

Figure 1: Inspirations for our models and training settings

39 **Gated recurrent-units as cortical circuits**    [3] highlights that Long short-term memories (LSTMs
40 [4]), the most commonly used gated recurrent units with memory cells controlled by input, output
41 and forget gates and these can be considered (at a high-level) as an abstraction of a cortical circuit
42 unit composed of layers of pyramidal cells with gating as shown in Figure 1a.

43 **Dopaminergic modulation of working memory**    As detailed in [5] and [6] and shown in Figure 1b,
44 the stimulus receive a reward from the dopaminergic nuclei based on which it is stored in the prefrontal
45 cortex (PFC) as persistent activations. Thus, biologically there is a reward-processing mechanism
46 (temporal difference, TD) that is operating outside the cortical memory.

47 **Deep Recurrent Reinforcement Learning**    As an extension of Deep Q-Networks (DQN)[7] with
48 recurrent units were proposed as DRQNs by [8] for partially observable environments. This model
49 naturally suits the non-Markov decision process involved in learning. ([9; 10; 11])

50 **Supervised Learning models**    We consider 2 different models for training recurrent networks to
51 output the probability of past occurrence of each image in the sequence using supervised learning:
52 LSTM and LSTM-Attention (LSTM-A). The attention augmented version ([12]) helps learn the
53 *context* better. These are used in conjunction with a CNN and trained using backpropagation through-
54 time (without truncation), as shown in Figure 2a.

55 **Reinforcement Learning models**    We consider an environment where each image in the sequence
56 is a state that the DRQN uses as input to approximate the Q-values of the two different action : 0
57 for *not seen* and 1 for *seen*. An $\epsilon$-greedy policy is learned using these Q-values to select the action
58 and receive a reward from the environment (positive value for correct action and negative value for
59 wrong action) as shown in Figure 2b. These are trained using common TD control algorithms such as
60 Q-learning and SARSA, indicated as Q+DQRN and SARSA+DQRN respectively.

## 3 Experiments and Results

62 For all experiments in both supervised and reinforcement learning settings, 100 images were drawn
63 from CIFAR-100 ([2]) dataset for each sequence. A CNN that was pre-trained on the same dataset

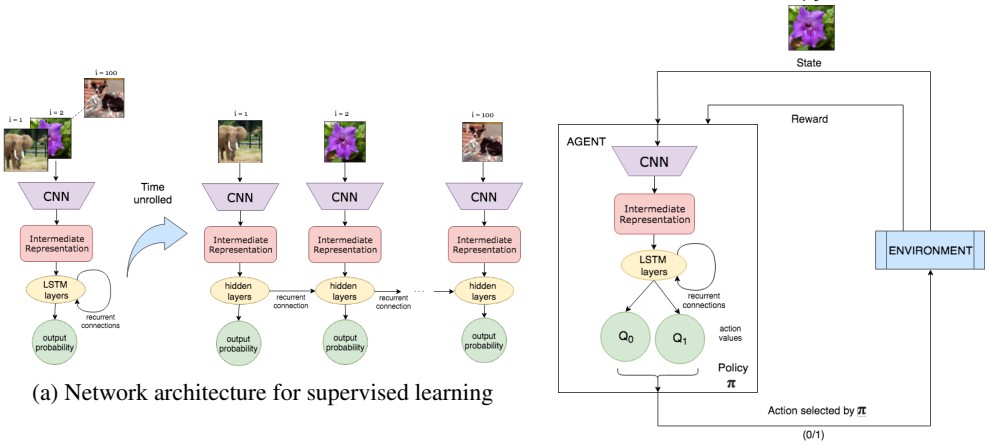

(a) Network architecture for supervised learning

(b) DRQN in an environment for RL

Figure 2: Model architectures

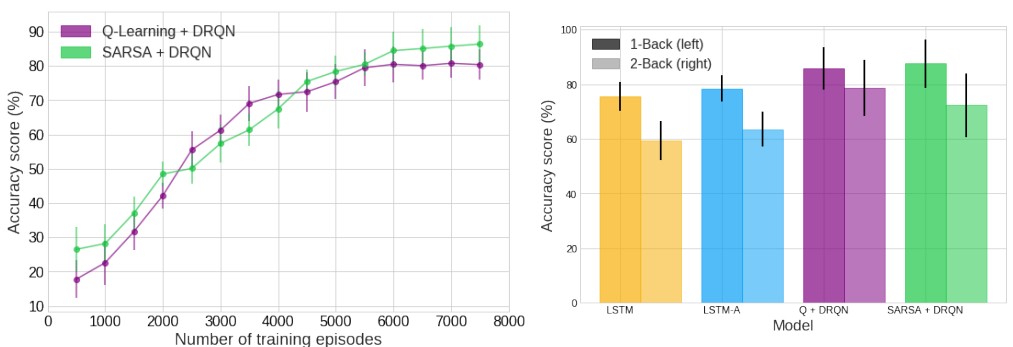

(a) Final test performance of DRQN models after every 500 training episodes (sequences)

(b) Final test performance of the 4 trained models for repeat counts $N = 1$ (left) and $N = 2$ (right)

Figure 3: Final performance of the models (the lines indicate std. dev.)

for the classification task was used for intermediate representation before LSTM layers (as shown in Figure 2).

The problem solved by all the models is a binary classification problem, predicting *unseen(0)/seen(1)*. The performance of all the models in the experiments were measured using the accuracy metric calculated based on the number of correct predictions for the 100 images in a sequence (as a %). This evaluation was repeated for 10 independent trials as a part of ablation studies.

Figure 3a shows the test accuracy of both the reinforcement learning models after fixed training episodes to track the progress. Figure 3b depicts the performance of the different models using mean final accuracy scores for two different conditions of repetitions of the stimuli images: $N = 1$ and $N = 2$. Figure 4 indicates the variation of test accuracy of each of the trained models over the course of a testing sequence. It can be seen that the models trained in the reinforcement learning setting outperform both the LSTM and LSTM-A in the supervised setting. Both on-policy and off-policy methods help in solving the task and learning good parametric functions in the LSTMs.

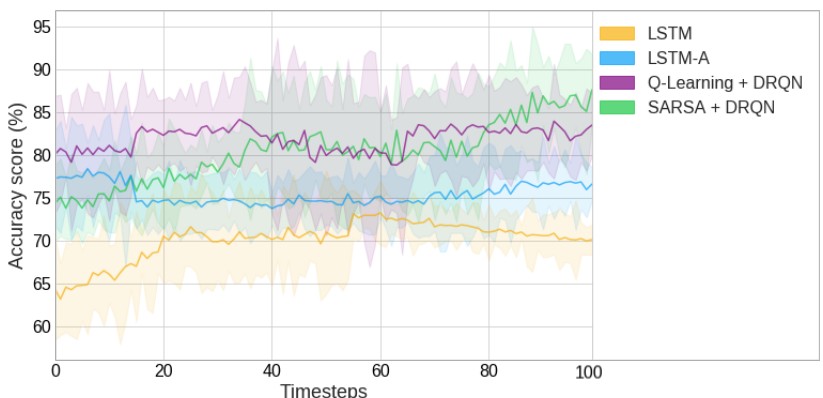

Figure 4: Performance of the trained models measured during test runs within a sequence

## 4   Conclusions and Future work

Popular literature in deep reinforcement learning ([8], [11]) outline many architectures and methods to solve specific (mostly motor) decision making tasks in an environment. In our work, we have taken an alternative view where we try to solve a strongly memory-oriented task using similar models used in reinforcement learning to emulate the reward processing happening to store memory.

A good working memory model understands what to remember and what to forget. From our study, we conclude that modelling a working memory using gated recurrent neural networks (such as LSTMs) to train using a reward-based learning approach offers significant advantages giving a reasonably superior performance. From the models studied,it can be seen that in addition to the capacity of the recurrent network, a training in the reinforcement learning setting offers a better generalization with its power for acting as a good computational model for the biological working memory for explaining the VWM task. This seems to indicate that the dopaminergic control of memory in the PFC is a high-level principle that is common to both artificial and biological neural systems. Also, this observation could be attributed to the fact that deep reinforcement learning is a better framework for learning in this case, given the non-stationary and non-Markov nature of the task involved.

As a next step, we plan to model other cognitive tasks for memory as a reinforcement learning problem and compare the performance of different algorithms and gating mechanisms in the networks. Further, we would like to extend our work by using the brain responses of humans solving the task from their fMRI data and identify the neural correlates of the visuo-temporal streams of information. This would throw more light on the functional similarities of the biological and artificial neural models explaining how the memory system functions computationally.

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
