# OpenReview forum: "Modelling Working Memory using Deep Recurrent Reinforcement Learning"
_NeurIPS.cc/2019/Workshop/Neuro_AI — Real Neurons & Hidden Units @ NeurIPS 2019 Poster_

### Official Review · AnonReviewer1 · 2019-09-23
**Interesting result, lacks composition**

**Clarity:** 2

**Category:**

AI->Neuro

**Clarity Comment:**

While the overall goal of the paper is clearly stated in the abstract, the publication loses clarity after the introduction. The "models" section is structured in an unusually segmented manner that fails to adequately detail the functional or structural similarities between them. Furthermore, the language use is poor. Overlapping clauses and run-on sentences dominate much of the text in this section.

The results section lacks a clear message comparing the performance of the different models. The figures show positive results regarding the ability of the models to perform a binary classification task on the CIFAR-100 image database, but the comparisons are incomplete. Moreover, the lack of any baseline comparison metric or statistical significance statements makes their importance hard to interpret. Figure captions themselves are hard to interpret, as they contain incomplete sentences and don't fully detail the information shown in the figure.

The biggest lack of clarity here is found in the gulf between motivation and result: if they're ostensibly attempting to model an organism's visual working memory functions, they haven't stated what organism that is and furthermore haven't made any real connection in the work between their presented models and the physiological systems that they're attempting to model. The only mention of biomimetic form or function is made in passing, and the reader is led to assume entirely the results of cited works by Braver and D'Ardenne. No confirmation or recreation of those cited results is attempted here.

**Evaluation:**

2: Poor

**Importance:**

2: Marginally important

**Importance Comment:**

The paper addresses the potential for modelling working visual memory processes with recurrent neural network architectures. Understanding these mechanisms is an important task in neuroscience, but I do not believe that the computational modeling approaches presented here are sufficient for publication.

**Intersection:**

3: Medium

**Intersection Comment:**

The presented models are clearly inspired by the structure and function of cortical visual stimulus processing regions, but that connection is at best one directional. No loop back to biological function is made in this paper regarding validation of their presented models. They state that the goal of this work is to apply these systems to the analysis of actual recorded data; such work may provide the full connection required to validate some aspects of the work presented here.

**Rigor Comment:**

The development of the models presented is not discussed and only references other work. Model implementation is also not discussed.

The results presented show evidence of systems learning more accurate representations of the labeled data that has been presented to them, but the differences between model performance metrics is not adequately explored here. Statistical significance statements are non-existent and no form of hypothesis test is presented regarding model accuracy. One can assume as much that a conjugate prior for the binary classification task can easily enough produce a 95% confidence statement regarding "chance" in this sort of test, but no such statement is presented.

**Technical Rigor:**

2: Marginally convincing

---

### Official Review · AnonReviewer3 · 2019-09-26
**Potentially important result but hard to assess rigor because it lacks detail.**

**Clarity:** 2

**Comment:**

One strength of the work is that the authors relate their models to biological models of working memory. However, the conclusions would be stronger if more details were included: 1) more details on the testing procedure, 2) what statistical test was used to arrive at the conclusions, and 3) what are the errorbars in Figures 2-4?

**Category:**

AI->Neuro

**Clarity Comment:**

The submission lacks clarity and detail.

**Evaluation:**

2: Poor

**Importance:**

3: Important

**Importance Comment:**

The authors test different mechanistic models of working memory. They conclude that a model trained with reinforcement learning outperforms the other models. However, the conclusions are hard to assess because the submission lacks detail.

**Intersection:**

4: High

**Intersection Comment:**

Testing different mechanistic models of working memory is important for both Neuro and AI.

**Rigor Comment:**

It is difficult to assess rigor.

**Technical Rigor:**

2: Marginally convincing

---

### Official Review · AnonReviewer2 · 2019-09-26
**Poor choice of task to study an interesting topic**

**Clarity:** 2

**Category:**

AI->Neuro

**Clarity Comment:**

Insufficient intuition for the results is presented. The authors are somewhat loose with their definition of a "context".

**Evaluation:**

2: Poor

**Importance:**

2: Marginally important

**Importance Comment:**

The paper focuses on working memory and reinforcement learning, which is an interesting topic. However, the choice of task is not a good one to probe this topic. The task as described is simply a familiarity detection task, and there is no reason to think that this is a good task for RL. In general, I do not see much that this study provides beyond previous work on deep RL.

**Intersection:**

3: Medium

**Intersection Comment:**

The general topic has the potential for interdisciplinary interest, but only if the task were changed to something more appropriate. The authors do not analyze the representations formed in the networks they study, which would be a necessary step to connect their approaches to biology.

**Rigor Comment:**

As described above, the choice of tasks is poor, which likely contributes to the modest improvement using RL as compared to supervised learning (it is not clear if this improvement in statistically significant). The paper does not provide intuition for the reason behind this purported improvement.

**Technical Rigor:**

2: Marginally convincing

---

### Decision · Program_Chairs · 2019-10-02

Accept (Poster)